# IHNV Infection Induces Strong Mucosal Immunity and Changes of Microbiota in Trout Intestine

**DOI:** 10.3390/v14081838

**Published:** 2022-08-22

**Authors:** Zhenyu Huang, Mengting Zhan, Gaofeng Cheng, Ruiqi Lin, Xue Zhai, Haiou Zheng, Qingchao Wang, Yongyao Yu, Zhen Xu

**Affiliations:** 1Department of Aquatic Animal Medicine, College of Fisheries, Huazhong Agricultural University, Wuhan 430070, China; 2State Key Laboratory of Freshwater Ecology and Biotechnology, Institute of Hydrobiology, Chinese Academy of Sciences, Wuhan 430072, China

**Keywords:** mucosal immune system, commensal microbiota, viral pathogens, RNA−seq, 16S rRNA sequencing

## Abstract

The fish intestinal mucosa is among the main sites through which environmental microorganisms interact with the host. Therefore, this tissue not only constitutes the first line of defense against pathogenic microorganisms but also plays a crucial role in commensal colonization. The interaction between the mucosal immune system, commensal microbiota, and viral pathogens has been extensively described in the mammalian intestine. However, very few studies have characterized these interactions in early vertebrates such as teleosts. In this study, rainbow trout (*Oncorhynchus mykiss*) was infected with infectious hematopoietic necrosis virus (IHNV) via a recently developed immersion method to explore the effects of viral infection on gut immunity and microbial community structure. IHNV successfully invaded the gut mucosa of trout, resulting in severe tissue damage, inflammation, and an increase in gut mucus. Moreover, viral infection triggered a strong innate and adaptive immune response in the gut, and RNA−seq analysis indicated that both antiviral and antibacterial immune pathways were induced, suggesting that the viral infection was accompanied by secondary bacterial infection. Furthermore, 16S rRNA sequencing also revealed that IHNV infection induced severe dysbiosis, which was characterized by large increases in the abundance of Bacteroidetes and pathobiont proliferation. Moreover, the fish that survived viral infection exhibited a reversal of tissue damage and inflammation, and their microbiome was restored to its pre−infection state. Our findings thus demonstrated that the relationships between the microbiota and gut immune system are highly sensitive to the physiological changes triggered by viral infection. Therefore, opportunistic bacterial infection must also be considered when developing strategies to control viral infection.

## 1. Introduction

The intestine of vertebrates is a primarily digestive organ that absorbs nutrients to ensure survival and reproduction but also constitutes one of the main entry points for pathogens. To fight these pathogens, gut−associated lymphoid tissues (GALTs) are the first immune line of defense and protect the host by initiating innate and adaptive immune responses. Thus, the gut of vertebrates also functions as a mucosal immune organ. Previous studies have demonstrated that teleost GALTs can recognize pathogens for elimination, although they lack the Peyer’s patches that are found in mammals [1]. Moreover, the mucosal immune system enables the colonization of mucosal surfaces by diverse microbial communities, thus providing many physiological, metabolic, and immunological benefits to the host. Meanwhile, commensal colonization contributes to maintaining the normal function of the immune system [2,3]. This cross-talk between the gut immune system and microbiota provides essential protection from pathogens. Therefore, additional studies are required to understand the complex interactions between the host, microbiota, and viruses.

A normal (i.e., beneficial) microbiota composition is critical for good health. Under normal circumstances, the symbiotic microbial composition is in homeostasis and the dominant microbial species in the fish gut remain consistent. However, this balance may be altered by various factors such as the diet, rearing conditions, and fish genotype. Previous studies have indicated that pathogen invasion including bacterial [4], viral [5], and parasitic infection [6] often disrupts microbial homeostasis. In mice, *Candida albicans* infection was associated with a loss of mucosal bacterial diversity with indigenous *Stenotrophomonas*, Alphaproteobacteria, and *Enterococcus* species dominating the small intestine mucosa [7]. Rotavirus infection alters the structure of the gut microbiota in neonatal calves [8]. Similarly, some studies have reported dysbiosis in the gut microbiota of teleost fish as a result of parasitic [9] or bacterial infection [10]. For instance, *Aeromonas hydrophila* infection alters the intestinal microbial community of zebrafish, in which δ−proteobacteria abundances increased from 0.33% to 18.6%, thus becoming the dominant bacteria [11]. However, the effects of viral infection on the teleost gut microbiota and intestinal immune system remain little known.

Infected with infectious hematopoietic necrosis virus (IHNV) is a fatal Novirhabdovirus that affects salmonid species including rainbow trout, Chinook salmon (*Oncorhynchus tshawytscha*), sockeye salmon (*Oncorhynchus nerka*), and Atlantic salmon (*Salmo salar*) worldwide [12]. Mortality is very high in juvenile fish, leading to significant losses in farmed fish populations [13]. Fish infected with this virus often present abdominal distension, bulging of the eyes, skin darkening, abnormal behavior, anemia, fading of the gills, and necrosis in the kidney and spleen [14]. Although we have previously demonstrated that IHNV can infect the digestive system and induce an immune response and dysbiosis in the oropharyngeal mucosa of adolescent rainbow trout [15], very little is still known regarding the pathogenesis of IHN disease in the juvenile trout gut, as well as its effects on innate and adaptive immune mechanisms and microbial community structure.

To assess the mucosal immune response and microbial community changes induced in the gut by IHNV, we infected juvenile rainbow trout via a developed immersion method. Our data indicated that IHNV infection can invade the gut mucosa, resulting in severe tissue damage, inflammation, and dysbiosis of the gut microbiota and eliciting a strong innate and adaptive immune response in the infected trout gut. Furthermore, transcriptome analysis demonstrated that not only antiviral but also antibacterial immune genes are upregulated in infected trout gut, suggesting that a secondary bacterial infection (manifested by an increase in the number and diversity of bacteria) is caused by the weakening of the organism by viral infection, and the increased amount of bacteria triggers an antibacterial immune response. Similar results have been described in common carp after spring viraemia of carp virus (SVCV) infection [16]. More importantly, the trout that survived after 28 days post-infection (DPI) exhibited almost no viral loads and their tissue and microbiome homeostasis was restored to pre−infection conditions. Therefore, our results provide insights into the interaction between the gut immune system, commensal microbiota, and viral pathogens.

## 2. Materials and Methods

### 2.1. Fish Maintenance

All experimental rainbow trout (average weights were 5 ± 1 g) were obtained from an aquatic farm in Chengdu (Sichuan province, China), and maintained and acclimatized in an aerated recirculating aquaculture system at 15 °C with an internal biofilter. Trout were fed with commercial trout pellets twice a day, and feeding was terminated 4 days before sacrifice. The trout pellets used in this study were provided by Skretting Aquaculture, which contains protein (37%), fat (23.8%), fiber (2.5%), and ash (6.3%). Animal procedures were approved by the Animal Experiment Committee of Huazhong Agricultural University and carried out according to the relative guidelines.

### 2.2. Infection of Fish with Infected with Infectious Hematopoietic Necrosis Virus (IHNV)

The epithelioma papulosum cyprini (EPC) cell line was maintained at 28 °C in a 5% CO_2_ atmosphere and maintained in minimum Eagle’s medium (MEM, Gibco, Gaithersburg, MD, USA) supplemented with 10% fetal bovine serum (FBS, Gibco, Amarillo, TX, USA), 100 mg/mL streptomycin, and 100 U/mL penicillin. The IHNV was propagated in EPC cells cultured in MEM medium with 2% FBS at 15 °C. After the extensive cytopathic effect (CPE), the EPC cells with IHNV were collected and repeated freezing and thawing three times for virus suspension. Subsequently, the supernatant was diluted proportionally (10^−1^~10^−12^) and inoculated into 96−well cell culture plates with 90% monolayer EPC cells at the bottom, and CPEs were observed every 24 h for 7 days to calculate the TCID_50_ of the IHNV. Then the dose of the IHNV was adjusted to 1 × 10^9^ pfu mL^−1^ in MEM and stored at −80 °C until use. For the infection experiment, trout were immersed with a dose of 6 mL IHNV (1 × 10^9^ pfu mL^−1^) in 10 L aeration water for 2 h at 15 °C. Then, trout were transferred to the aquarium containing new aquatic water. As a control, the same number of trout were same treated with the MEM. Then, tissue samples were taken after 1, 4, 7, 14, 21, and 28 DPI.

### 2.3. Sample Collection

For sampling, we anesthetized rainbow trout with MS−222, and collected tissues at the shown time points after infection. For the detection of viral loads and the study of immune−related gene expression, the head kidney and gut of trout for all time points were collected in sterile micro−centrifuge tubes. For histology and pathology study, a segment of gut was clipped and immediately fixed in 4% (*v*/*v*) neutral−buffered paraformaldehyde for at least 24 h. For 16S rRNA sequencing, gut samples from 4, 14, and 28 days post−infected fish and control fish were collected in sterile freezing tubes. These tissues collected for RNA or DNA analyses were immediately frozen with liquid nitrogen and stored at −80 °C for further study.

### 2.4. RNA Isolation and Quantitative Real−Time PCR (qRT−PCR) Analysis

Total RNA was extracted from various tissues using a TRIzol Reagent (Invitrogen, Carlsbad, CA, USA) according to the manufacturer’s protocol. The concentration of extracted RNA was determined by spectrophotometry (Nanodrop ND1000, LabTech, Holliston, MA, USA), and the integrity of the RNA was determined by 1% agarose gel electrophoresis (Agilent Bioanalyser, 2100). To normalize gene expression levels for each sample, equivalent amounts of total RNA (1000 ng) were used for cDNA synthesis with the SuperScript first−strand synthesis system in a 20 μL reaction volume. The resultant cDNA was stored at −20 °C. To detect the abundance of IHNV, specific primers were used and shown in Appendix A. The relative expressions of the immune−related genes were determined by qRT−PCR with each specific primer (Appendix A). qRT−PCR was conducted using the MonAmp^TM^ SYBR Green qPCR Mix (Monad, Suzhou, China) following the manufacturer’s instructions. The internal control gene elongation factor 1α (EF1α) was employed as a reference gene. Relative mRNA abundances were calculated using the 2 ^−ΔΔCt^ method and normalized to EF1α. Viral loads were determined by constructing an IHNV plasmid standard curve, and average values from duplicates of each gene in the samples were extrapolated using the standard curve to calculate the IHNV copy numbers. All data were expressed as the mean ± standard error estimate (SEM). The student’s *t*−test was conducted using GraphPad (veision 9.0, Harvey Motulsky, CA, USA).

### 2.5. Histology, Light Microscopy, and Immunofluorescence Studies

To assess the morphological changes in the gut after infection of IHNV, histopathological examination was used in this experiment. Briefly, after being dissected, the gut of rainbow trout was fixed in 4% neutral−buffered formalin overnight at 4 °C and then transferred to graded ethanol for dehydration and dimethyl benzene for vitrification. After that, the tissues were embedded in paraffin and cut into 5 μm−thick sections with a rotary microtome (MICROM International GmbH, Walldorf, Germany). After being stained with conventional hematoxylin and eosin (HE) or Alcian blue (AB), the sections were examined under the microscope (Olympus, BX53, Shinjuku City, Tokyo, Japan) using the Axiovision software to acquire and analyze images. For the detection of IHNV−infected cells in the gut tissue, the paraffin sections were stained with mouse anti−IHNV−*N* mAb (mouse IgG for isotype; 1 μg/mL; BIO−X Diagnostics, Rochefort, Belgium) at 4 °C overnight. As controls, the mouse IgG was used at the same concentration. After washing three times with PBS, Cy3−conjugated AffiniPure goat anti−mouse IgG pAb (3 μg/mL) was added and incubated at RT for 40 min. All sections were stained with DAPI (4′, 6−diamidino−2 phenylindole; 1 μg/mL; Invitrogen, Carlsbad, CA, USA) for 8 min before mounting. All images were acquired and analyzed using an Olympus BX53 fluorescence microscope (Olympus, Shinjuku City, Tokyo, Japan) and the iVision−Mac scientific imaging processing software (Olympus, Shinjuku City, Tokyo, Japan).

### 2.6. RNA−Seq Library Construction, Sequencing, and Data Analyses

The gut samples of the control group and the IHNV−infected group of 4 and 14 DPI were sent to Seqhealth Technology Co., Ltd. (Wuhan, China). Briefly, total RNA was extracted using a TRIzol reagent (Invitrogen, Carlsbad, CA, USA) and was used for stranded RNA sequencing library preparation using a KCTM Stranded mRNA Library Prep Kit for Illumina ^®^ following the manufacturer’s instruction. PCR products corresponding to 200–500 bps were enriched, quantified, and finally sequenced on a Hiseq X 10 sequencer (Illumina, San Diego, CA, USA). Reads were mapped to the *Oncorhynchus mykiss* genome using Spliced Transcripts Alignment to a Reference (STAR) (version 2.5.3a, developed by Alexander Dobin et al.) with default parameters. The mapped reads were analyzed via feature counts. Differential expression genes were estimated by the edgeR package [17]. The genes with low expression (CPM (counts per million) < 1 in three or more samples) were excluded from downstream analysis. The resulting genes were considered as differentially expressed genes (DEGs) if FDR ≤ 0.05 and |log_2_ (fold−change) | ≥ 1. For further analysis of the DEGs, we carried out a Kyoto encyclopedia of genes and genomes (KEGG) enrichment to identify the immune−related pathways that were significantly enriched following viral infection.

### 2.7. Bacterial 16S rRNA Sequencing and Data Analyses

Purified amplicons were pooled in equimolar and paired−end sequenced (2 × 300) on an Illumina MiSeq platform with MiSeq Reagent Kit v3 at Shanghai Personal Biotechnology Co., Ltd. (Shanghai, China). Microbiome bioinformatics was performed with QIIME2 [18] with slight modification according to the official tutorials (https://docs.qiime2.org/2022.2/ (accessed on 11 May 2022)). Briefly, raw sequence data were demultiplexed using the demux plugin followed by primers cutting with the cutadapt plugin [19]. Sequences were then quality filtered, denoised, merged, and chimera removed using the DADA2 plugin [20]. Alpha−diversity metrics (Chao1 and Shannon and, beta−diversity metrics (weighted UniFrac) were estimated using the diversity plugin. Taxonomy was assigned to ASVs using the classify−sklearn naïve Bayes taxonomy classifier in the feature classifier plugin against the SILVA (Release 132, Bremen, Germany). For Lefse analysis, the non−parametric factor Kruskal–Wallis rank sum test was applied for determining the species that showed significant differences in abundance. By linear discrimination analysis (LDA), the effect of the different species was estimated.

## 3. Results

### 3.1. Construction of Infected with Infectious Hematopoietic Necrosis Virus (IHNV) Infection Model

To evaluate the gut immune responses in juvenile trout under viral infection, we developed a bath infection model with IHNV, and nine fish were collected at each sampling point (Figure 1a). The fish were closely monitored each day and 30% of the infected fish died within 15 days (Figure 1b). At 4 DPI, typical clinical signs appeared in the infected trout, including darkening of the skin, pale gills, exophthalmia, petechial hemorrhages, and empty gut (Figure 1c). qPCR analyses revealed that the IHNV loads began to increase in the trout head kidney and gut at 4 DPI, and these high levels persisted at 7, 14, and 21 DPI (Figure 1d,e). Moreover, using an anti−IHNV−*N* monoclonal antibody (mAb) and immunofluorescence microscopy, we detected obvious viral fluorescence signals in the lamina propria and epidermis of the gut villus in 4 DPI fish (Figure 1f). These results indicated that IHNV had successfully invaded the fish gut mucosa. Moreover, we incubated EPC cells with infected trout gut homogenate supernatants, and CPEs were clearly observed when compared with the controls (Figure 1g), indicating that the viruses in the fish gut were still highly virulent.

### 3.2. Histopathological Changes and Immune Gene Expressions in the Gut

H&E staining was then conducted to assess the morphological changes in fish gut mucosal tissue after viral infection. In 7 DPI fish, we identified some cell shedding around the gut villus, and the length−width ratios of gut villi in infected fish were significantly decreased compared to that of control trout (Figure 2a,c). Moreover, using AB staining, we found that the number of mucous cells in the gut villi was significantly increased at 7 and 14 DPI compared to the control groups (Figure 2b,d). We further evaluated the expression levels of different immune−related genes in the fish gut at all sampling time points via qPCR analysis. The expression levels of innate immune genes (CCL19, IL−8, and IRF8) and antiviral genes (STAT, IFNAR, MDA5, Vig1, and Mx1) were higher at 4 and 7 DPI, whereas adaptive immune genes (IgT, IgM, and CD22) were mainly expressed at 14 and 28 DPI (Figure 2e). These results further suggested that the rainbow trout was successfully invaded by IHNV, and this virus triggered strong immune responses. 

### 3.3. Analysis of Transcriptomic Changes in Trout Gut after IHNV Infection 

To further study the immune responses in the trout gut upon IHNV infection, RNA−seq was conducted on the Illumina platform, and samples collected at 4 and 14 DPI were compared to the control group. A total of 632,406,846 paired−end raw reads were generated, from which 592,972,312 high−quality reads were obtained after filtration. Next, unique mapped reads were further filtered and differential gene expression analyses were only conducted when there were more than 10 reads for a given gene in three or more individual libraries.

Upon IHNV infection, the mRNA expression of 478 genes was found to be significantly altered (426 genes upregulated and 52 genes downregulated) at 4 DPI, whereas 4128 genes were dysregulated (2596 genes upregulated and 1532 genes downregulated) at 14 DPI (Figure 3a,b). Based on the transcriptome data, KEGG pathway enrichment analysis was performed to investigate the functions of the identified DEGs among the three groups. Our findings revealed that the upregulated DEGs were highly associated with key signaling pathways including the Toll−like receptor signaling pathway and RIG−I−like receptor signaling pathway at 4 DPI (Figure 3c). These two crucial signaling pathways play an important role in recognizing pathogens and triggering innate immune responses [21]. Additionally, we found that several DEGs in the 14 DPI group were enriched in pathways involved in the regulation of multiple cellular physiological processes such as the PI3K−Akt signaling pathway, which plays an important role in cell growth and proliferation, and the Rap 1 signaling pathway, which controls cell adhesion and participates in cell–cell junction formation (Figure 3d) [22,23]. These results revealed different biological processes in the trout gut that contribute to the response to IHNV infection at different time points. Moreover, we detected many significantly upregulated antiviral genes in both the 4 and 14 DPI groups (Figure 3e), which was consistent with the above−described qPCR results. Interestingly, we also detected high expression levels of antibacterial genes (SAA, CHTA−2B, CMPK2, CD209, LECT2, and NK−lysin) by RNA−seq (Figure 3f), suggesting that viral infection disrupts the microbial balance of the gut.

### 3.4. IHNV Infection Results in Gut Microbial Dysbiosis 

16S rRNA sequencing of gut samples from control and infected fish was conducted to investigate the effects of IHNV infection on the abundance and diversity of trout gut microbiota communities. A total of 2,390,474 raw reads were obtained from the control and IHNV−infected fish samples. After filtering the raw reads using DADA2, 1,702,694 high−quality clean reads were obtained, and these reads were used for downstream analyses. Afterward, the sequences were divided into unique amplicon sequence variants (ASVs) at a 97% similarity threshold using the DADA2 plugin. We obtained 16,777 ASVs, and the total numbers of ASVs detected in the experimental groups (4 DPI: 6896 ASVs; 14 DPI: 5623 ASVs; 28 DPI: 1893 ASVs) were much higher than that in the control group (2365 ASVs), indicating that IHNV infection altered the microbial communities in the fish gut.

Next, we analyzed the differences in microbial diversity and community in the trout gut between the control and IHNV−infected fish. Four indices were used to measure alpha−diversity, including community richness (Chao1) and community diversity (Shannon). Interestingly, compared to the control fish, the Chao1 and Shannon indices were all significantly higher in the fish gut microbial communities at 4 and 14 DPI (Figure 4a,b). In contrast, we did not identify any obvious changes in alpha−diversity in the gut microbiota of the 28 DPI fish (Figure 4a,b). Additionally, using weighted UniFrac distances, we calculated the beta−diversity of the control, 4, 14, and 28 DPI samples (Figure 4c). Principal coordinates analysis (PCoA) revealed that the gut microbiota of the control fish was distinct from that of the 4 and 14 DPI fish but not the 28 DPI fish. Specifically, our findings indicated that the community richness and diversity of the gut microbiota increased at 4 and 14 DPI but then returned to normal levels at 28 DPI.

To analyze the bacterial composition changes in IHNV−infected individuals, the microbial sequences from the control and infected fish were classified at the phylum, class, order, family, and genus levels. For all detected samples, the predominant bacterial phyla were Proteobacteria, Firmicutes, Actinobacteria, and Bacteroidetes (Figure 4d). Compared to the control fish, the 4 and 14 DPI fish exhibited a decrease in the relative abundances of Proteobacteria (66.4% in the control group versus 24.8% and 33.3% in the 4 and 14 DPI fish) and Actinobacteria (15.9% in the control group versus 10.9% and 11.4% in the 4 and 14 DPI fish) (Figure 4d). Notably, the Firmicutes and Bacteroidetes abundances in the 4 DPI (41.7% and 34.4%) and 14 DPI fish (11.7% and 9.7%) were significantly higher compared to the controls (9.8% and 1.2%). In the 28 DPI fish, the microbial communities appeared to recover. Then, we analyzed the bacterial composition changes at the order level. Compared to the control fish, the orders *Clostridiales*, *Bacillales*, and *Bacteroidales* exhibited higher abundances in the 4 and 14 DPI fish, whereas the abundances of *Vibrionales* and *Actinomycetales* decreased (Figure 4e). These data further demonstrated that IHNV infection could induce microbial dysbiosis in fish gut mucosal sites. LDA effect size (LEfSe) analysis was conducted to further explore the microbial biomarkers that contributed to the changes in bacterial structure in the different groups. In the 4 and 14 DPI fish, we found that the abundances of the *Lachnospiraceae*, *Ruminococcaceae*, *Bacteroidaceae*, and *Bacillaceae* families were significantly increased by more than 4−fold. In contrast, the family *Microbacteriaceae* was decreased by more than 4−fold (Figure 4f). In the 28 DPI fish, only the family *Moraxellaceae* was found to increase more than 4−fold, whereas the abundances of five families (*Vibrionaceae*, *Halomonadaceae*, *Rickettsiaceae*, *Pseudonocardiaceae*, and *Streptococcaceae*) decreased more than 4−fold (Figure 4f).

### 3.5. Bacterial Community Changes in Different Groups at The Genus Level 

Using heat map analysis, we compared the top 50 bacterial genera whose abundances were affected in the four fish groups and found that the abundances of most bacteria increased at 4 DPI (Figure 5a). Notably, we found that several potentially pathogenic genera were significantly increased at 4 and 14 DPI compared to the control group, including *Bacteroides*, *Prevotella*, *Alistipes*, and *Shigella* (Figure 5b). Moreover, several beneficial genera also exhibited increased abundances, including *Faecalibacterium*, *Bacillus*, *Clostridium*, and *Bifidobacterium* (Figure 5c). In contrast, the abundance of several pathogenic and beneficial bacteria such as *Halomonas*, *Paracoccus*, *Vibrio*, and *Streptococcus* was lower in the control fish (Figure 5d).

## 4. Discussion

The intestinal mucosal surfaces of vertebrates are inhabited by complex and diverse microbial communities and are also continuously exposed to a wide variety of pathogens. During the evolution of vertebrates, the fish gut has developed a complete mucosal immune system with a unique array of innate and adaptive immune cells and molecules to tolerate commensals while fighting pathogens [24]. However, the three−way interactions between pathogens, microbiota, and teleost hosts are complex and poorly understood. Therefore, our study sought to evaluate the relationships between the microbiota and gut immune system during viral infection.

Studies have demonstrated that the juvenile stage of salmonid life is the most vulnerable to being infected with infectious hematopoietic necrosis virus (IHNV), and older fish have more resistance to clinical disease [13]. Therefore, the juvenile rainbow trout were selected for the viral infection experiment. Here, we developed a natural infection model by immersing rainbow trout in an IHNV−containing bath. Notably, the 4 DPI fish exhibited pseudofaeces, in addition to other typical clinical symptoms of IHN disease including abnormal swimming behavior, pale gills, petechial hemorrhages around the eyes, and darkening of the skin [25]. qPCR and immunofluorescence analysis showed high IHNV RNA copies in the gut tissue of diseased trout and the virus was detected in the gut villus, indicating that IHNV had successfully invaded the intestine. Additionally, we observed severe tissue damage, inflammation, and decreased length–width ratios of the trout gut villus after IHNV infection, which might be due to the shedding of intestinal cells. In line with these morphologic changes and viral loads, the trout gut mucosa exhibited a significant upregulation of antiviral genes (STAT1, MDA5, IFNAR, Vig1, and Mx1), which are known to be essential for the production of effective antiviral responses in the mammalian intestinal mucosal surfaces, suggesting that viral invasion had effectively triggered an immune response in the intestine. More specifically, IHNV challenge increased the expression of genes associated with innate immunity (e.g., chemokine ligand, interleukin, complement factor, and interferon regulatory factor) at 4 and 7 DPI, whereas genes associated with adaptive immunity (e.g., IgT, IgM, and IgD) were upregulated after 14 DPI. These findings suggest that the innate and adaptive responses mainly exerted function during the early and late stages of infection, respectively. Nevertheless, it is not clear whether these local immune responses were specifically triggered by the induced IHNV infection or some opportunistic bacteria. To confirm this, additional transcriptome analyses were conducted using the 4 and 14 DPI gut samples and our findings indicated that the 14 DPI group had a much higher number of DEGs than the 4 DPI group, suggesting that the gut response induced by IHNV at 14 DPI was extremely strong. Next, we analyzed the expressions of different types of immune genes at 4 and 14 DPI. As expected, IHNV infection activated several antiviral immune pathways and upregulated many virus−specific immune genes. At 4 DPI, several significantly upregulated genes were found to be mainly enriched in the RLR and TLR signaling pathways. Both of these pathways are classified as pattern recognition receptor (PRR) pathways, which are well conserved across all vertebrates [26,27,28]. RLRs are a family of cytosolic pattern recognition receptors that are essential for detecting viral RNA and initiating the innate immune response. The RLR family includes three members: retinoic acid−inducible gene I (RIG−I), melanoma differentiation−associated gene 5 (MDA5), and laboratory of genetics and physiology 2 (LGP2) [29,30]. In this study, we detected high expression levels of two RLRs (i.e., LGP2 and MDA5) in infected trout. Upon viral recognition, MDA5 interacts with mitochondrial antiviral−signaling protein (MAVS), thus inducing the activation of downstream signaling pathways. Importantly, we detected a significant upregulation of IRF−3 and IRF−7 in the PRR pathways. IRF−3 and IRF−7 are primary transcriptional factors downstream of the MAVS signaling pathway, which regulate the type I IFN response after RNA virus infection [31,32]. This process ultimately leads to the expression of multiple antiviral genes (e.g., Vig1, Mx2, and IFNG). At 14 DPI, the majority of the significantly upregulated genes were enriched in the PI3K−Akt signaling pathway. In higher vertebrates, PI3K/Akt signaling represents a two−faced player in interactions between the virus and cell, able to both promote viral replication and actively participate in the immune response that ultimately inhibits viral replication [33]. Several studies have characterized the interaction between viruses and the PI3K−Akt signaling pathway in fish [34,35]. However, additional studies are needed to elucidate the specific mechanisms through which the PI3K−Akt signaling pathway promotes or inhibits viral replication. The strong antiviral immune responses found at 4 DPI were similar to our previous studies in trout swim bladder mucosa after infection by IHNV [36]. Interestingly, a potent induction of IFN and Mx gene expression has been detected in Pleuronectiformes upon challenge with hirame novirhabdovirus and viral hemorrhagic septicemia virus [37]. Additionally, SVCV infection induced both RLR and TLR pathways activation at mucosal sites in common carp [38]. Differently, there was no significant change in TLR expression in trout skin after infection with parasite, while almost all the significantly different genes in the complement pathway were upregulated, suggesting that the complement components play a crucial role in antiparasitic invasion [39]. Besides, many genes of antibacterial peptides and proinflammatory cytokines were significantly expressed in fish skin and gill under bacterial infection [40,41]. Thus, all these results suggested that the immune responses in fish mucosal tissues triggered by different pathogens were not the same. Moreover, we found that the cytokine–cytokine receptor interaction pathway was significantly enriched in both the 4 and 14 DPI fish. Previous studies have reported that cytokine signaling played crucial roles in the immune response to bacterial infection [42]. Additionally, several antibacterial genes (e.g., SAA, CATH−2B, LECT2) were significantly upregulated in infected trout compared to the control. Our findings thus indicated that IHNV infection not only triggered an antiviral response in the intestinal mucosa but weakened the organism, which promoted the commensal bacteria to turn into opportunists or pathogenic and resulted in a secondary infection that triggered a strong antibacterial immune response.

Next, we identified the changes in the microbiome composition in the trout gut upon viral infection. Notably, compared to the controls, the diversity of the gut microbiota composition was significantly increased at 4 and 14 DPI, indicating that viral invasion disturbed the gut microbiome. Similar to previous studies in the fish gut, Proteobacteria was the most abundant phylum in the control group [15,43]. After viral infection, the abundances of Proteobacteria decreased at 4 and 14 DPI. A similar phenomenon has been observed in fish skin and pharynx upon pathogen infection [38,39]. Interestingly, Dong et al. reported that no significant changes in Proteobacteria abundances were detected in the adolescent trout (10–15 g) gut after IHNV infection, which indicated that IHNV is more likely to invade juvenile rainbow trout [15]. In contrast, the abundance of Proteobacteria often increases in mammals in response to disease [44,45]. These discrepancies might be attributed to variations in the functional roles of microbes in different animal species. It is also worth noting that the abundances of the Bacteroidetes and Firmicutes phyla were significantly increased at 4 and 14 DPI. In mammals, the Firmicutes/Bacteroidetes (F/B) ratio is widely believed to have an important influence in maintaining normal intestinal homeostasis. An increase or decrease in the F/B ratio is regarded as dysbiosis, where the former is commonly associated with obesity and the latter with inflammatory bowel disease (IBD) [46]. In our study, the F/B ratio in the control fish was 8.05 but decreased to 3.56 and 3.53 in 4 and 14 DPI fish, respectively. These findings indicated that enteritis occurred in the trout gut at 4 and 14 DPI, which is consistent with the transcript levels of inflammation−associated genes. Next, LEfSe and heat map analyses provided further insights into the differences in the gut microbiota of the control and infected fish. In this study, the largest number of biomarkers were concentrated at 4 DPI. Notably, the abundance of *Bacteroides* was >5%, which had been demonstrated to be associated with inflammatory bowel disease in humans [47]. Similarly, in grouper iridovirus−infected fish, the abundance of *Bacteroides* has also been reported to increase compared to the control group [48]. Moreover, the abundance of several other genera (e.g., *Prevotella*, *Alistipes*, and *Shigella*) was significantly increased at 4 DPI. In humans, the three genera are harmful and able to cause inflammation and many severe diseases [49,50,51,52,53,54]. Previous studies certificated that the abundance of *Alistipes* in the gut of juvenile large yellow croaker significantly increased under adverse environmental conditions [55]. A study in *Epinephelus coioides* found that the proportion of *Escherichia Shigella* increased in the intestine of nervous necrosis virus and *Vibrio*−infected groups [48]. Moreover, the potentially pathogenic bacteria abundances increased in the skin of infected fish with parasite and caused secondary bacterial infection [39]. Thus, it might be a commonality that the abundances of pathogenic bacteria would increase in mucosal tissues in diseased fish. Interestingly, infected fish also exhibited increased abundances of some commensal and beneficial bacteria. *Faecalibacterium prausnitzii* is the sole known species of the genus *Faecalibacterium*, which produces butyrate and other short−chain fatty acids through the fermentation of dietary fiber [56]. Many strains of *Bacillus* are currently used as probiotic dietary supplements in humans and animals, including *B. coagulans*, *B. clausii*, *B. cereus*, *B. subtilis*, and *B. licheniformis* [57]. Several studies have shown the ability of probiotic *Bacillus* to enhance the immunity of fish to withstand the pathogenicity of virus [58,59]. The abundances of these two genera increased significantly at 4 DPI by more than 4%. Additionally, increased abundances of other beneficial genera (e.g., *Clostridium* and *Bifidobacterium*) were detected in 4 DPI fish [60,61]. Stabilized fermentation product of *Cetobacterium somerae* improves gut and liver health and antiviral immunity of zebrafish [62]. *Clostridium butyricum* could protect against the infection of Carassius auratus herpesvirus in gibel carp [63]. Our results indicated that the increased bacterial diversity in 4 and 14 DPI fish was due to higher abundances of both beneficial and pathogenic bacteria. However, we also detected decreases in the abundance of both beneficial (e.g., *Halomonas* and *Paracoccus*) and harmful bacteria (e.g., *Vibrio* and *Streptococcus*) in the infected group [64,65,66,67]. Overall, our results showed that IHNV infection disturbed the balance of the gut microbiome in fish, which promoted the colonization of both beneficial and pathogenic bacteria and changed the overall structure of the bacterial communities.

At 28 DPI, the abundances of predominant bacteria at the phylum level (Proteobacteria, Actinobacteria, Firmicutes, and Bacteroidetes) recovered considerably. However, there were still some differences in the abundances of individual phyla, suggesting that the surviving trout intestine itself can maintain bacterial homeostasis. Similarly, in common carp infected with SVCV, the abundances of Firmicutes and Proteobacteria in the pharynx were observed to increase and decrease at 4 DPI compared to control fish, respectively, and then recovered to normal levels at 28 DPI [38]. Although several reports showed that fish tissue microbiota changed when fish was in a severe diseased state infected by bacteria and parasite [39,48], studies involved in the microbial composition of survived fish after pathogenic infection were scarce. One review summarized the mechanisms involved in the restoration of homeostasis, including bacterial co−aggregation, production of biosurfactants, antimicrobials, and signaling molecules that target the host or pathogens, competitive exclusion of pathogens, immunomodulation, and factors that increase tight junction barrier function on the host epithelium [68]. However, additional studies are needed to explore the mechanisms involved in microbiome recovery in fish.

In conclusion, our study provides insights into the interactions between viral infection, host gut immune responses, and bacterial community composition. IHNV infection could induce dramatic immune responses in the trout gut. In the early stage of infection, PRR pathways were activated and the innate immune response and expression of antiviral genes were significantly increased, suggesting that the innate immune response played an important role in this stage. After 14 DPI, a marked upregulation of the adaptive immune gene IgT was detected. Our latest study demonstrated the participation of a specific IgT in viral neutralization in the swimming bladder of fish [36]. However, additional studies are needed to determine whether viral infection induces the production of specific IgT in the gut. On the other hand, viral invasion induced serious pathological changes, resulting in microbial dysbiosis, which was most prominently characterized by a decrease in the abundance of Proteobacteria in the trout gut. As expected, our study found that the abundance of pathogenic bacteria significantly increased after viral infection, indicating that the damage to the gut villus induced by IHNV infection might cause secondary bacterial infection. Overall, our findings demonstrated that IHNV−infected trout exhibited a state of dysbiosis in the gut, resulting in the proliferation of pathobionts and invasion of the gut mucosa, which in turn led to secondary bacterial infection. Future studies are warranted to determine whether there are similar results in other growth stages, including the older stage when fish are less susceptible to IHNV infection and the younger stage when fish have not yet developed a robust intestinal immune system. Trout GALTs can elicit immune responses in the gut to neutralize viruses and restore microbial homeostasis. Interestingly, our findings demonstrated that the abundance of some beneficial bacteria also increased significantly after infection, which might play a role in resisting multiple infections. Therefore, additional studies are needed to explore the relationship between the teleost gut beneficial microbiota and specific immune responses in the trout gut against IHNV infection.

## Figures and Tables

**Figure 1 viruses-14-01838-f001:**
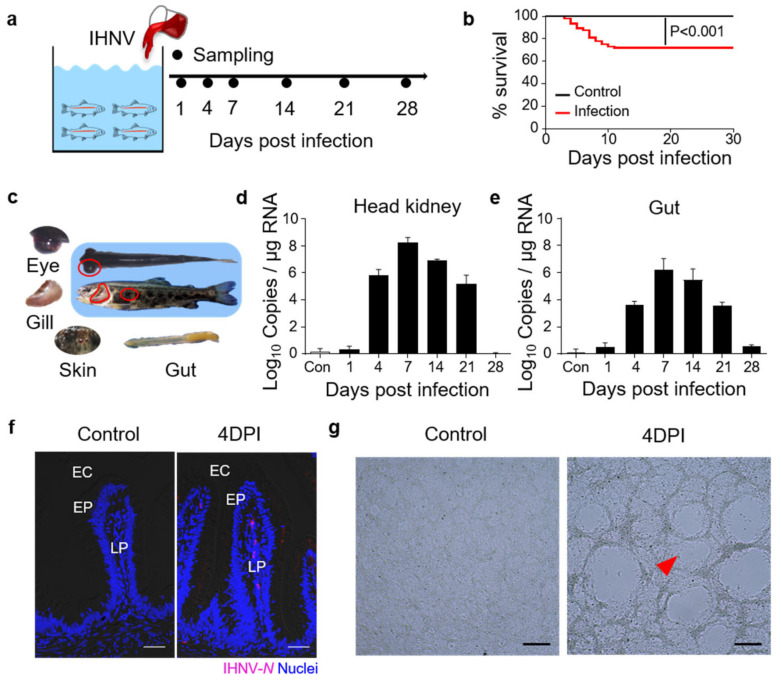
Construction of the model of infected with infectious hematopoietic necrosis virus (IHNV)−infected rainbow trout. (**a**) Scheme of the infection strategy with IHNV by bath. (**b**) Cumulative survival rate of control and IHNV−infected fish. (**c**) The clinical signs of IHNV−infected trout at 4 DPI with darkening of the skin, pale gills, exophthalmia, petechial hemorrhages, and empty gut. (**d**,**e**) IHNV−*G* gene copies (Log_10_) were quantified using qPCR in fish tissues collected at 1, 4, 7, 14, 21, and 28 DPI. The virus was detected in the fish head kidney (**d**), and gut (**e**). (**f**) Immunofluorescence staining of IHNV in gut paraffin sections from control and 4 DPI fish (*n* = 9). IHNV (red) is stained with an anti−IHNV−*N* mAb; nuclei are stained with DAPI (blue). EC, enteric cavity; EP, epidermis; LP, lamina propria. Scale bars, 50 μm. (**g**) Cytopathic effect of IHNV on EPC cells after cultured with the supernatant of gut homogenates from control and the 4 DPI fish. Scale bars, 100 μm. Data in (**d**,**e**) are representative of at least three independent experiments (mean ± SEM).

**Figure 2 viruses-14-01838-f002:**
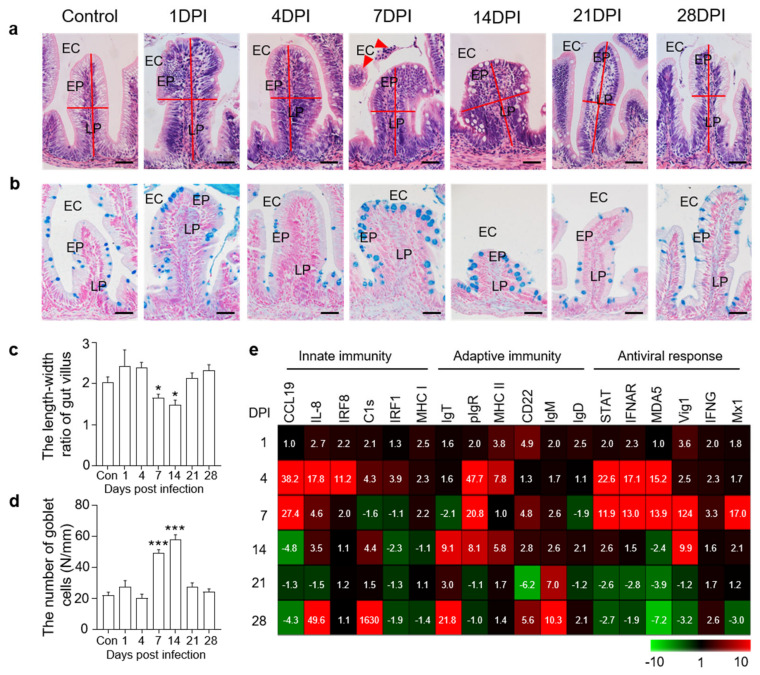
Histopathological changes and immune gene expressions in the trout gut after IHNV infection. (**a**) H&E staining of the gut from control and experimental fish infected with IHNV after 1, 4, 7, 14, 21, and 28 days (*n* = 6 fish per group). EC, enteric cavity; EP, epidermis; LP, lamina propria. Scale bars, 50 μm. The red line indicates the length or width of gut villus. The red triangle indicates shedding cells. (**b**) AB staining of the gut from control fish and experimental fish infected with IHNV after 1, 4, 7, 14, 21, and 28 days (*n* = 6 fish per group). EC, enteric cavity; EP, epidermis; LP, lamina propria. Scale bars, 50 μm. (**c**) The length−width ratio of gut villus of control and infected fish (*n* = 6 fish per group). (**d**) The number of goblet cells of gut villus from control and infected fish (*n* = 6 fish per group). (**e**) Heat maps illustrate results from quantitative real−time PCR of transcripts for selected immune markers from gut of IHNV−infected versus control fish measured at 1, 4, 7, 14, 21, and 28 DPI (*n* = 6 fish per group). Data are expressed as mean fold increase in expression. Color value: fold change. Statistical differences were evaluated by unpaired Student’s *t*−test. Data in (**c**,**d**) are representative of six independent experiments (mean ± SEM). (*) *p* < 0.05, (***) *p* < 0.001.

**Figure 3 viruses-14-01838-f003:**
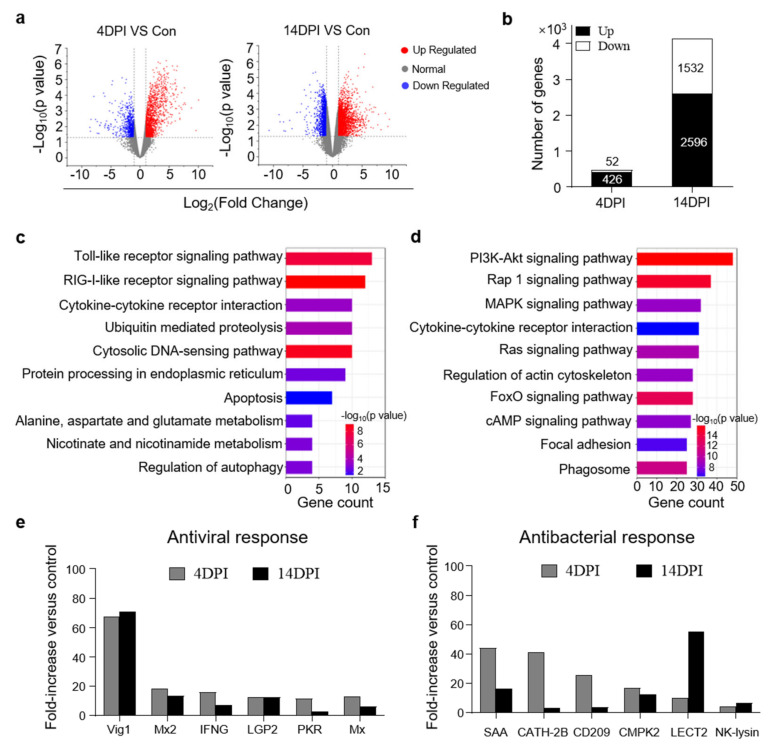
Transcriptome analyses of trout gut upon IHNV infection. (**a**) Volcano plot showing the overlap of genes upregulated or downregulated in the gut of rainbow trout at days 4 (left) and 14 (right) after infection with IHNV−infected versus control fish. Red spots are expression of fold change > 2 and FDR < 0.05; blue spots are expression of fold change < 2 and FDR < 0.05; grey spots mean no difference in expression. (**b**) The counts of upregulated or downregulated genes in the gut of rainbow trout at 4 or 14 DPI versus control fish. (**c**,**d**) KEGG pathways were significantly altered in gut of rainbow trout at 4 and 14 DPI versus control fish revealed by RNA−seq studies. (**e**,**f**) Representative antiviral (**e**) and antibacterial (**f**) response genes modulated by IHNV infection at 4 and 14 DPI. Fold change differences between control and IHNV−infected groups were calculated using cutoff of twofold.

**Figure 4 viruses-14-01838-f004:**
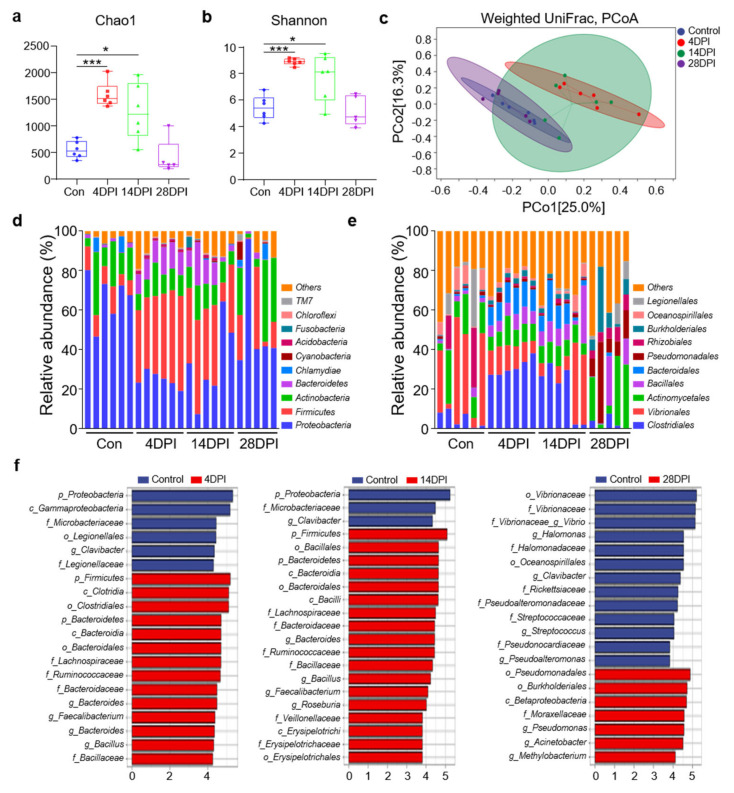
Changes in the composition and abundance of trout gut microbiota community in response to IHNV infection. (**a**,**b**) Richness and diversity of bacterial community in trout gut from control and infected groups. Richness and diversity of the gut bacterial community were measured using Chao1 and Shannon index. Error bars represent standard error of mean (SEM). (*) *p* < 0.05, (***) *p* < 0.001 (unpaired Student’s *t*−test). (**c**) Weighted UniFrac compositional−based distances were computed for control (blue−colored), 4 DPI (red−colored), 14 DPI (green−colored), and 28 DPI (pink−colored) groups, respectively. The principal coordinate analysis (PCoA) results are presented as two−dimensional ordination plots, which were generated using two (PCo1 and PCo2) principal coordinates. (**d**,**e**) Composition and relative abundance of the top 10 dominant bacterial taxa in the gut of rainbow trout in control fish and IHNV−infected fish at 4, 14, and 28 DPI at the phylum (**d**) and order (**e**) levels. (**f**) LEfSe cladogram of differentially abundant taxa in control and IHNV−infected fish at 4, 14, and 28 DPI (phylum (p), class (c), order (o), family (f), genera (g)).

**Figure 5 viruses-14-01838-f005:**
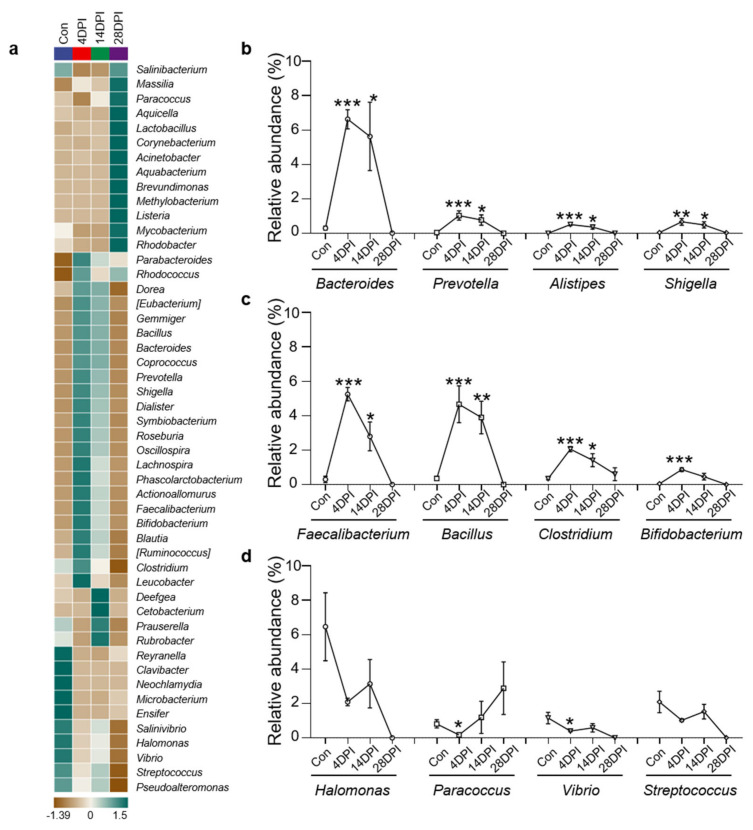
Changes of gut bacterial genera in response to IHNV infection. (**a**) Heatmap compared the composition of the top 50 genera with average abundance in different groups in the gut of rainbow trout. (**b**) Relative abundance of *Bacteroides*, *Prevotella*, *Alistipes*, and *Shigella* in the gut of rainbow trout in control and IHNV−infected groups of 4, 14, and 28 DPI. (**c**) Relative abundance of *Faecalibacterium*, *Bacillus*, *Clostridium*, and *Bifidobacterium* in the gut of rainbow trout in control and IHNV−infected groups of 4, 14, and 28 DPI. (**d**) Relative abundance of *Halomonas*, *Paracoccus*, *Vibrio*, and *Streptococcus* in the gut of rainbow trout in control and IHNV−infected groups of 4, 14, and 28 DPI. (*) *p* < 0.05, (**) *p* < 0.01, (***) *p* < 0.001 (unpaired Student’s *t*−test).

## Data Availability

Not applicable.

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
