# Peer review of "IHNV Infection Induces Strong Mucosal Immunity and Changes of Microbiota in Trout Intestine"

_viruses, 2022, doi:10.3390/v14081838_

Round 1

Reviewer 1 Report

The first thing to point out is that the title of this article does not attract readers’ interest. I found that the author has studied many indicators in this research, but not all indicators must appear in the title. 

The scientific name of the species should be fully defined.

Keywords should be expressed as capital for the first letter.

Line 66 and elsewhere: please define the IHNV  for the first use.  

The authors should describe any measures taken to avoid these errors (e.g. sample size) and/or discuss the limitations of their study.

Line 94: please refer to the commercial feed information (e.g., source, composition, ...etc)

Author Response

We would like to thank the reviewers for their insightful and constructive comments. We have addressed all the comments and attached them in one document. In this document, the newly added and corrected text is in bold with red color.

Reviewer 2 Report

The submitted manuscript  presents a very well planned and carried out study on changes of intestinal immunity and commensal microbiota diversity in juvenile rainbow trout during IHNV infection. The viral infection induced a strong innate and adaptive immune response in the gut, and RNA-Seq analysis showed that both antiviral and antimicrobial immune pathways were induced, suggesting that the viral infection was accompanied by a secondary bacterial infection. Relationships between microbiota and the intestinal immune system are very sensitive to physiological changes caused by viral infection. Therefore, opportunistic bacterial infections must also be considered when developing strategies to control viral infection. The results obtained by authors provide insight into the interactions between the gut immune system, commensal microbiota, and viral pathogen in IHNV infected trout. In conclusion, the presented results are very valuable research material due to the fact that they concern a very important commercial fish species, which is widely distributed all over the world.

I consider this article important for the described field of fish immunology and recommend it for publication, but I have some comments and suggestions for the authors that may improve its quality.

 Line 81-84, 425-427

Upregulation of antimicrobial genes in trout infected with IHNV does not imply that the viral infection elicited a strong antimicrobial immune response. This is too much of a simplification. It only means that a secondary bacterial infection (manifested by an increase in the number and diversity of bacteria) is caused by the weakening of the organism by viral infection. The increased amount of bacteria triggers an antibacterial immune response. In many viral infections in fish, secondary bacterial infections result in increased mortality after eradication of the virus. Please consider changing these lines in order not to suggest misleading statements to your readers.

Paragraph 2.1 and 2.2, 3.1 and Figure 1a.

The text describes the model of IHNV infection, which is then again described in paragraph 3.1 and in Figure 1a. It would be clearer for the reader to describe the infection model in the Material and methods section and include Figure 1a there. Now, this is in my opinion unnecessarily separated and described in 3 sections and in the description to Figure 1a. For example, it is only in the description of the drawing that the reader learns how many fish were collected at each point.

Line 375

„Diarrhea”

The authors do not seem to be fully aware of the clinical course of IHNV during e.g natural infection, because they say they have observed “diarrhea” in infected fish. In the course of IHNV infection, as in other rhabdovirus infections in salmonid fish, which starts through the gills and digestive tract, one of the characteristic symptoms is the exfoliation of intestinal epithelial cells, as can be seen in Fig. 2 a and b, where the process was captured in 7dpi. This is probably a nomenclature misunderstanding, because it has been known for many years that many infections, including IHNV but also other rhabdoviruses, occur in fish with catarrhal enteritis which results in the formation of stringy feces / mucus cylinders (pseudofaeces). If the authors had this symptom in mind, the term “diarrhea” was used inaccurately and could be changed to “pseudofaeces”.

Paragraph 4. Discussion

Transcriptomic responses in the fish intestine:

The results presented are very well compiled and are a valuable contribution to the discipline, but would be much more useful to readers interested on fish immunity if discussed against the background of other fish findings.

It would be good to consider the work of other authors on, for example, infection with another fish virus or on fish immunity in organs other than the intestine. The pattern of triggering the immune response is similar also in the course of parasitic or bacterial infections in fish. It is worth remembering that in rhabdovirus infection it is the juvenile rainbow trout that is the most vulnerable form due to the fact that they do not yet have a well-developed adaptive immunity.

Gut microbiome:

The presented results on gut microbiome in trout are a very important study of the topic salmonids diseases. However, there are several other reports on gut microbiome in fish, also salmonids, e.g. with bacterial, parasitic or viral infections. It would be worthwhile to discuss the results against the background of fish reports, which are somewhat available in the literature, instead of mentioning the article on gut recovery in patients with Covid infection, because that adds nothing. There are even reports that prebiotics or probiotics have a positive effect on minimizing the course of viral infection by regulating intestine microbiome in fish. Instead of citing reports on terrestrial vertebrates, it would be worth focusing on aquatic animals, as this would raise the rank of the results obtained by authors.

Author Response

We are pleased to resubmit the revised version of the manuscript entitled "IHNV infection induces strong mucosal immunity and changes of microbiota in trout intestine” by Zhenyu Huang et al., which we wish to be considered for publication in Viruses. We would like to thank the editors and reviewers for their insightful and constructive comments. We have addressed all the comments and attached them in one document. In this document, the newly added and corrected text is in bold with red color.

Round 2

Reviewer 1 Report

The paper is greatly improved and can be accepted in the current form.